# The Use of Diatomite as a Catalyst Carrier for the Synthesis of Carbon Nanotubes

**DOI:** 10.3390/nano12111817

**Published:** 2022-05-26

**Authors:** Meruyert Nazhipkyzy, Renata R. Nemkayeva, Araylim Nurgain, Aigerim R. Seitkazinova, Balaussa K. Dinistanova, Almagul T. Issanbekova, Nurzhamal Zhylybayeva, Nurgul S. Bergeneva, Gulnar U. Mamatova

**Affiliations:** 1Department of Chemical Physics and Material Science, Al-Farabi Kazakh National University, Almaty 050040, Kazakhstan; nurgain93@mail.ru (A.N.); aikolove00@mail.ru (A.R.S.); b.dinistanova@gmail.com (B.K.D.); mamatovag@mail.ru (G.U.M.); 2Institute of Combustion Problems, Almaty 050012, Kazakhstan; nurzhamaljk@mail.ru; 3National Nanotechnology Laboratory of Open Type, Al-Farabi Kazakh National University, Almaty 050040, Kazakhstan; quasisensus@mail.ru; 4Department of Geography and Environmental Sciences, Al-Farabi Kazakh National University, Almaty 050040, Kazakhstan; almagul_83_83@mail.ru (A.T.I.); nurgul.bergeneva@kaznu.kz (N.S.B.)

**Keywords:** diatomite, carbon nanotubes, catalyst, chemical vapor deposition, propane-butane gas mixture

## Abstract

In this article, multiwalled carbon nanotubes (MWCNTs) have been synthesized on the surface of a diatomite mineral impregnated with transition metal salts using a propane-butane mixture in a chemical vapor deposition reactor at atmospheric pressure. The catalyst concentration and synthesis temperature have been varied in order to understand their effects on the formation of MWCNTs and their morphology. Diatomite was chosen as a catalyst carrier due to its elemental composition. It is mainly composed of amorphous silica, quartz and also contains such metals as Fe, K, Ca, Mn, Cr, Ti, and Zn, which makes it a promising material for use as a catalyst carrier when synthesizing carbon nanotubes (CNTs) by catalytic chemical vapor deposition (C-CVD). For the synthesis of carbon nanotubes by C-CVD on the surface of the diatomite, the following salts were used as a catalyst: CoCl_2_·6H_2_O; Ni(NO_3_)_2_·6H_2_O, and the concentrations of the solutions were 0.5; 1.0 and 1.5 M. Natural diatomite was characterized by X-ray diffraction analysis (XRD) and Scanning Electron Microscopy (SEM) analysis.

## 1. Introduction

Carbon nanotubes were discovered by Iijima in 1991 a little bit after the discovery of fullerenes. Since then, due to their properties, studies on the synthesis of CNTs have not lost their relevance.

The traditional method for the synthesis of carbon nanotubes is the sputtering of carbon by laser radiation or in an electric arc in the presence of nanocatalysts [1,2,3,4]. Despite numerous studies in this direction, the optimization and development of new strategies to synthesize CNTs are still relevant [5,6].

It has been stated that the rational choice for a catalyst plays an important role in controlling the morphology, yield and quality of MWCNTs in connection with the synthesis process [7].

Fe-, Co-, Ni-based metallocenes appeared to be especially promising for achieving a yield and a good quality of MWCNTs. The catalyst is the key factor in the synthesis of CNT when using CVD methods [8,9,10,11].

When comparing several methods utilized for the synthesis of carbon nanotubes, catalytic chemical vapor deposition is found to be the most amenable method for large-scale production [12,13]. C-CVD can be operated under relatively mild conditions and at atmospheric pressure, which makes the process both economically and technologically attractive [14,15].

Recently, nanostructured materials have attracted enormous interest in the field of adsorption with their extraordinary adsorption capacity and their property of being easily recycled. Consequently, CNTs, including both the multi-walled types and the single-walled types, have been widely employed for this purpose [16,17]. They show a great advantage in the removal of contaminants, especially heavy metal ions, in wastewater, in construction [18] and in the field of energy storage systems.

Diatomaceous earth is often used as a catalyst carrier and as a wastewater treatment agent due to its high permeability, high porosity and chemical inertness. In [19], diatomite-CNT composites were used as a promising and highly efficient adsorbent for the removal of phenolic compounds from wastewater. Meanwhile, the composite (diatomite and SWCNT) obtained by the authors of the work [20] had superparamagnetic properties and was used as a magnetic separation sorbent for the adsorption of metal ions from aqueous media.

In this work, multi-walled carbon nanotubes (MWCNTs) were synthesized on the surface of natural diatomite by chemical vapor deposition (CVD) method. The distinguishing feature of this work from the previous works is that the obtained MWCNT on the surface of diatomite is not subjected to modification. In addition, heterogeneous systems have a lower cost and higher efficiency compared to expensive pure CNTs with a better adsorption performance.

Thus, MWCNTs synthesized on the surface of natural diatomite could prospectively be used as an adsorbent of heavy metals, such as lead, cadmium, zinc, etc., in wastewater treatment.

## 2. Materials and Methods

The general scheme of the setup for the synthesis of CNT by the method of catalytic decomposition of hydrocarbon vapors is shown in Figure 1 [21].

For the production of carbon nanotubes by the C-CVD method, the diatomite mineral from the Mugaldzhar deposit (Aktobe region, Kazakhstan), presented in Figure 2, was used as a catalyst carrier.

From the SEM analysis of natural diatomite, one can clearly see that it has a porous structure.

We prepared two solutions with three different concentrations (0.5, 1.0 and 1.5 M) of Co(NO_3_)_2_·6H_2_O; Ni(NO_3_)_2_·6H_2_O, respectively. Then, 0.5 g of diatomaceous earth was impregnated with an alcohol solution of Co(NO_3_)_2_·6H_2_O; Ni(NO_3_)_2_·6H_2_O, respectively, and dried for 13–15 min at a temperature of 80–100 °C, necessary and sufficient for the evaporation of water and alcohol. As a result, a metal-containing catalyst (diatomite) was obtained in the form of a powder. Then, it was placed in a reactor and heated in an argon flow to 400–500 °C. In order to synthesize carbon nanotubes, the propane-butane mixture served as a carbon carrier due to its availability.

Then, the temperature for the synthesis of carbon nanotubes was increased (650 °C, 700 °C, 750 °C and 800 °C), and a propane-butane gas mixture was supplied for 30 min at a flow rate of 90 cm^3^/min. Then, the reactor was cooled to room temperature in argon for 1–1.5 h.

The synthesis of CNTs requires the formation of primary product nuclei and the formation of a metal-carbon interface [22]. We will give an example for a nickel catalyst:(1)$$\mathrm{Ni}{\left({\mathrm{NO}}_{3}\right)}_{2}\cdot 6H_{2}O\overset{300\;{}^{\circ}C}{\to }\mathrm{NiO}+2{\mathrm{NO}}_{2}↑+0.5O_{2}↑+6H_{2}O$$
(2)$$2\mathrm{NiO}+C\overset{200–400\;{}^{\circ}C}{\to }2\mathrm{Ni}+CO_{2}↑$$

Due to the diatomite impregnation of diatomite with an alcohol solution of transition metal salts and drying, catalysts with active centers in the form of nickel were obtained. Nickel-containing catalysts are most catalytically active with respect to hydrocarbon gas mixtures during their pyrolysis in the synthesis of carbon nanotubes. The activity of nickel is conditioned by the temperature range of stability of the existence of carbide phases in Ni-C systems [23]. According to this concept, the pyrolysis of a propane-butane mixture on nickel-containing catalysts proceeds as the decomposition of propane-butane, followed by the formation of nickel carbide and the growth of CNTs.

Ethyl alcohol volatilizes under standard conditions with the formation of a catalytic active component—nickel, cobalt.

The mechanism of CNT formation is realized as the decomposition of a propane-butane mixture on the surface of a metal nanoparticle, the formation of a carbide on the surface, the diffusion of carbon into the bulk of the crystal, and the deposition of carbon on the surface of a metal nanoparticle. The growth of multi-walled and single-walled nanotubes on Ni catalysts proceeds by the root mechanism, when the catalyst particles remain on the surface of the substrate or carrier.

The role of the catalyst is reduced to the adsorption of the initial carbon-containing compound on the surface of the catalyst particle, dissociation of this compound, dissolution of carbon in the volume of the catalyst particle, and subsequent release of dissolved carbon to form single-walled, multi-walled nanotubes or nanofibers [24].

The obtained samples were studied by scanning electron microscopy on Quanta 3D 200i (FEI company, Hillsboro, OR, USA) and Raman spectroscopy on Solver Spectrum (NT-MDT, Moscow, Russia) using a 473 nm laser.

## 3. Results

Figure 3 represents the X-ray diffraction pattern of natural diatomite, which shows that diatomite contains quartz (a polymorphic modification of silicon dioxide).

On the X-ray pattern of the natural diatomite, the most characteristic line of quartz with a distance of 3.35 Å is visible, and there are several lines that characterize amorphous SiO_2_ with inter-planar distances of 2.49 and 4.26 Å, as well as lines with inter-planar distances of 3.03, 3.28 and 3.39 Å, which indicate the presence of calcite in the sample.

The components of diatomite are also present: Illite-montmorillonite, Al_2_(Si_2_O_5_)(OH)_4_ (kaolinite) are present in small quantities. SiO_2_ (quartz) is present in a somewhat larger amount; in addition, a substantial amount of X-ray amorphous phase is present.

Typical Raman spectra of carbon samples obtained by C-CVD at synthesis temperatures of 650 °C, 700 °C, 750 °C and 800 °C and a maximal catalyst concentration of 1.5 M are shown in Figure 4.

Raman spectra of all the studied samples are represented by two main characteristic carbon peaks—G in the region of 1570–1600 cm^−1^ and D at ~1360 cm^−1^. The width of these peaks can be used for the assessment of the crystallinity degree of carbon materials. Therefore, a detailed analysis of Raman spectra was performed with respect to the FWHM (full width at half maximum) of the D and G peaks at various synthesis temperatures and catalyst concentrations (Figure 5 and Figure 6).

For all concentrations of the Ni catalyst, a decrease in FWHM(D) is observed with an increase in the synthesis temperature. As can be seen from Figure 4a, at high synthesis temperatures of 750 °C and 800 °C and all catalyst concentrations, the spectra demonstrate similarly low values of FWHM(D), close to typical crystal peak widths.

At the same time, it is interesting that in the case of samples obtained on the basis of a Co catalyst, an explicit temperature dependence is not observed and their Raman spectra demonstrate relatively low values of FWHM(D) for almost all synthesis parameters—the temperature and catalyst concentrations.

The width of the main graphite peak G also demonstrates a different dependence on the synthesis temperature for carbon nanotubes based on Ni and Co catalysts (Figure 6a,b).

In the case of the Ni catalyst, FWHM(G) at first declines and then starts to grow with an increasing synthesis temperature (Figure 6a). This can be explained by the fact that when the temperature changes from 700 to 800 °C, along with an increase in the crystallinity of the structure, a peak D’ begins to appear as a shoulder in the region of 1610 cm^−1^, which is responsible for finer structural defects in crystalline sp^2^ materials, contributing to the peak width G.

As for carbon nanostructures based on the Co catalyst, there is no certain correlation of FWHM(G) with the synthesis temperature for 0.5 M and 1 M concentrations. On the other hand, for a 1.5 M concentration, which provides a more uniform distribution of the catalyst over diatomite, one can observe a decline of the G peak width with an increasing synthesis temperature, thus indicating an increase of the crystalline phase in the forming carbon structures.

Figure 7a,b shows the dependences of the intensity ratios I(D)/I(G) on the synthesis temperature for different molar masses of the catalysts. In highly disordered carbon materials, the intensity ratio of D and G peaks is generally less than one (I(D)/I(G) < 1). The increase in the structural order leads to the growth of this parameter until the sizes of ordered areas reach ~ 10–20 nm (I(D)/I(G) ≥ 1). Further growth of crystallinity, in the case of systems with sizes of ordered areas above 20 nm, leads to the decrease of I(D)/I(G). Therefore, taking into account the values of FWHM of D and G peaks, a steady growth of I(D)/I(G) with the synthesis temperature indicates a decrease in defects and, conversely, an increase in the crystalline order in carbon structures based on the Ni catalyst (Figure 7a).

The I(D)/I(G) ratio in Raman spectra of carbon nanotubes synthesized on diatomite with the 0.5 M Co catalyst (Figure 7b) demonstrates a downward trend, which indicates a decrease in the number of defects and/or amorphous phase in the structure of the formed CNTs with an increase in the synthesis temperature. For concentrations of 1 and 1.5 M, this parameter fluctuates—after a sharp drop at 750 °C, an increase is observed at 800 °C. Nevertheless, Raman spectra of samples obtained at high temperatures and the 1.5 M Co catalyst show I(D)/I(G) values in the range of 0.3–0.7, which corresponds to CNTs with fewer defects and a therefore higher structural quality.

The intensity of the 2D band in the region of 2600–2800 cm^−1^ is usually used for the characterization of MWCNTs. The dependences of the ratio of intensities I(G)/I(2D) on the synthesis temperature are shown in Figure 8a,b.

The graphs shown in Figure 8a for CNTs synthesized on the Ni catalyst characterize an increase in the long-range order of the structure with an increase in the temperature. The lower the value of I(G)/I(2D), the higher the long range order. It can be seen that in the case of the Co catalyst, the long-range order remains high at all concentrations and temperatures, thus indicating a high quality of the formed CNTs (Figure 8b).

A detailed analysis of Raman spectra allows one to conclude that at synthesis temperatures of 650 °C and 700 °C, for all Ni catalyst concentrations, the resultant samples correspond to amorphous carbon (a-C) or carbon nanotubes of low quality (LQ CNT). Meanwhile, samples obtained at 750–800 °C using 1M and 1.5M of the Ni catalyst can be attributed to MWCNTs of medium quality (MQ CNT). In the case of the Co catalyst, we can conclude that almost all considered synthesis parameters lead to the formation of MWCNT of various qualities.

Figure 9a,b illustrates CNTs’ quality, where we tried to provide a generalizing visual representation of the results of the Raman spectra analysis. The quality of nanotubes was assessed by summing up such parameters as FWHM(G), I(D)/I(G), and I(G)/I(2D). The increasing degree of quality is shown by the arrow on the corresponding axis, so that it varies from a-C (amorphous carbon) to HQ CNT (high-quality carbon nanotubes).

Thus, the effect of the reagents and synthesis conditions on the properties of synthesized nanostructured materials, such as carbon nanotubes, was studied. Additionally, the degree of disorder in the obtained multi-walled carbon nanotubes was determined from the Raman spectra.

It was revealed that the optimal conditions for the C-CVD process for the growth of high-quality carbon nanotubes consisted in the use of a 1.5 M solution of cobalt at temperatures of 750 and 800 °C. Figure 10 shows the SEM images of MWCNTs, demonstrating the best quality according to Raman spectroscopy. According to the SEM measurements, the diameter of the obtained carbon nanotubes ranges from 30 nm to 170 nm.

## 4. Conclusions

In this work, the synthesis of carbon nanotubes on a diatomite surface with various catalysts was carried out. Natural diatomite was investigated by XRD and SEM. Their characteristic features are the presence of amorphous active silicon dioxide on the one hand and a fine-porous structure, lightness, and low thermal conductivity on the other. These properties make these materials chemically highly active and make it possible to use them as sorbents, catalysts, filtering and heat-insulating materials, and filler carriers. According to the results of the Raman spectroscopy, the growth of multi-walled carbon nanotubes was observed on selected catalysts, such as Co and Ni, at temperatures of 750 and 800 °C.

In addition, heterogeneous systems have a lower cost and higher efficiency compared to expensive pure CNTs with a better adsorption performance.

It was stated that the optimal conditions for the C-CVD process for the growth of high-quality carbon nanotubes were: a 1.5 M solution of CoCl_2_·6H_2_O at temperatures of 750 and 800 °C.

## Figures and Tables

**Figure 1 nanomaterials-12-01817-f001:**
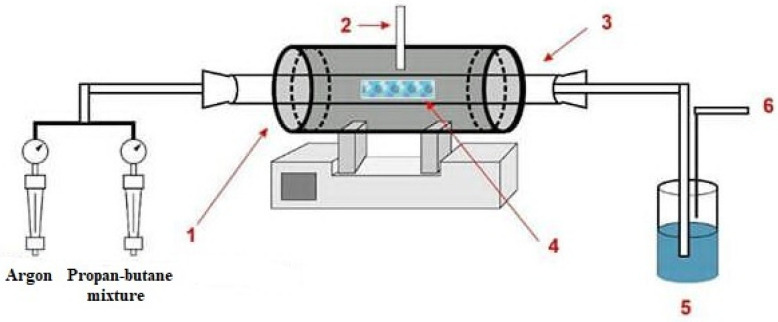
Schematic representation of an experimental setup for the synthesis of MWCNTs. 1—furnace; 2—thermocouple; 3—quarts tube; 4—catalyst; 5—controller of gas; 6—emitted smoke.

**Figure 2 nanomaterials-12-01817-f002:**
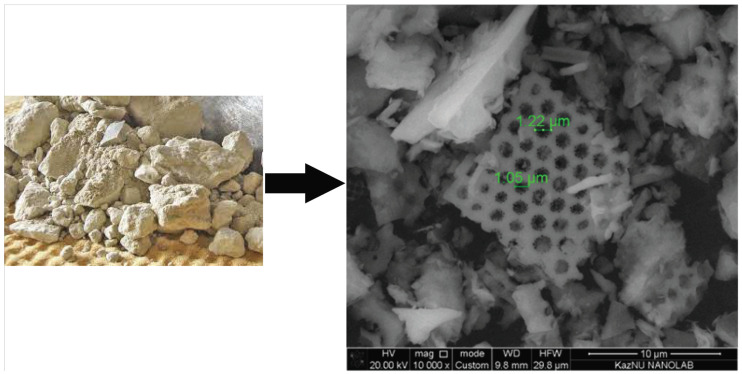
Natural diatomite (Aktobe region, Kazakhstan) and its SEM analysis.

**Figure 3 nanomaterials-12-01817-f003:**
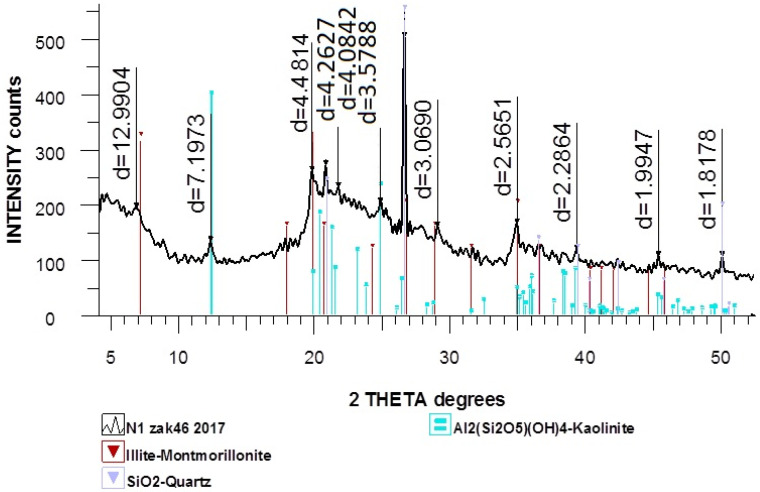
X-ray diffraction pattern of natural diatomite.

**Figure 4 nanomaterials-12-01817-f004:**
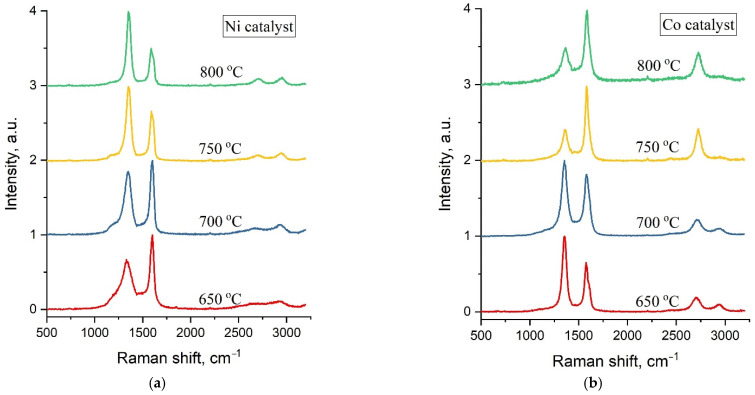
Raman spectra of carbon samples synthesized using (**a**) Ni and (**b**) Co catalysts at different temperatures.

**Figure 5 nanomaterials-12-01817-f005:**
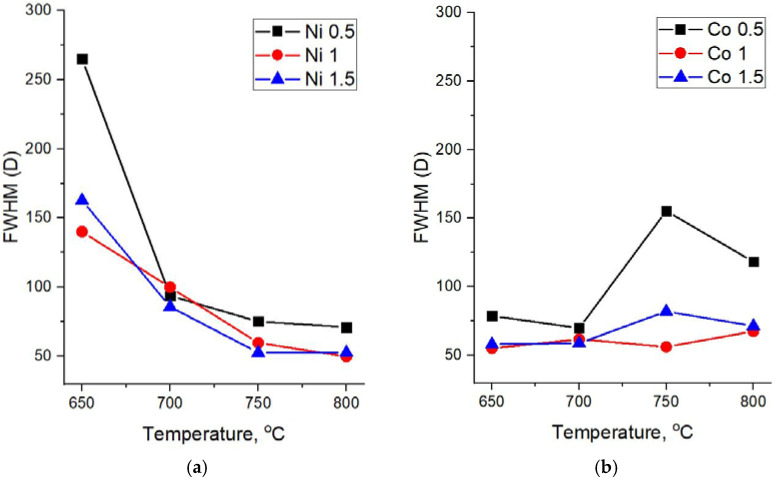
Dependence of the FWHM of the D peak on the synthesis temperature. (**a**) Ni and (**b**) Co catalysts.

**Figure 6 nanomaterials-12-01817-f006:**
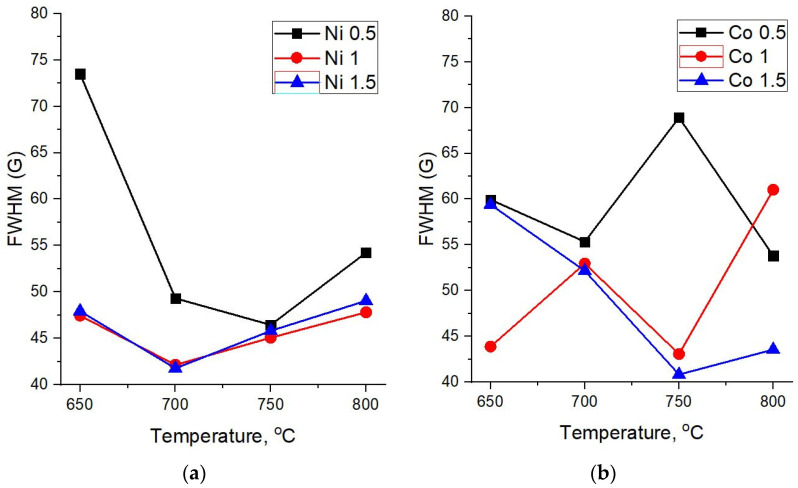
Dependence of the G peak width on the synthesis temperature of MWCNTs. (**a**) Ni and (**b**) Co catalysts.

**Figure 7 nanomaterials-12-01817-f007:**
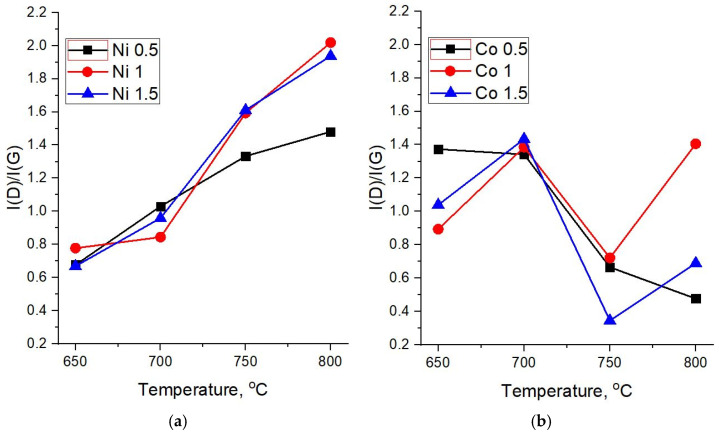
Dependence of the ratio of intensities I(D)/I(G) on the synthesis temperature of MWCNTs. (**a**) Ni and (**b**) Co catalysts.

**Figure 8 nanomaterials-12-01817-f008:**
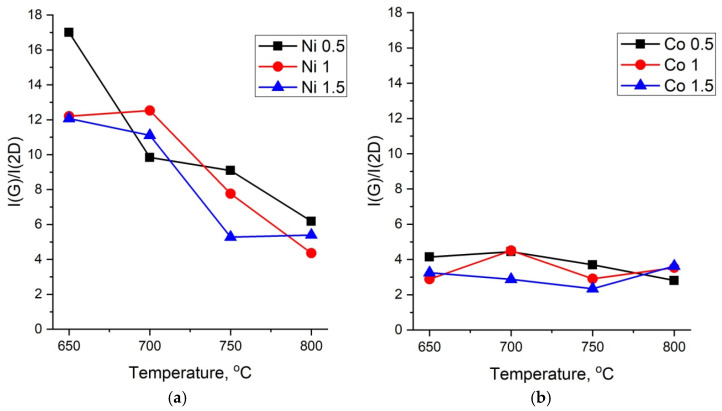
Dependences of the ratio of intensities I(G)/I(2D) on the synthesis temperature of MWCNTs. (**a**) Ni and (**b**) Co catalysts.

**Figure 9 nanomaterials-12-01817-f009:**
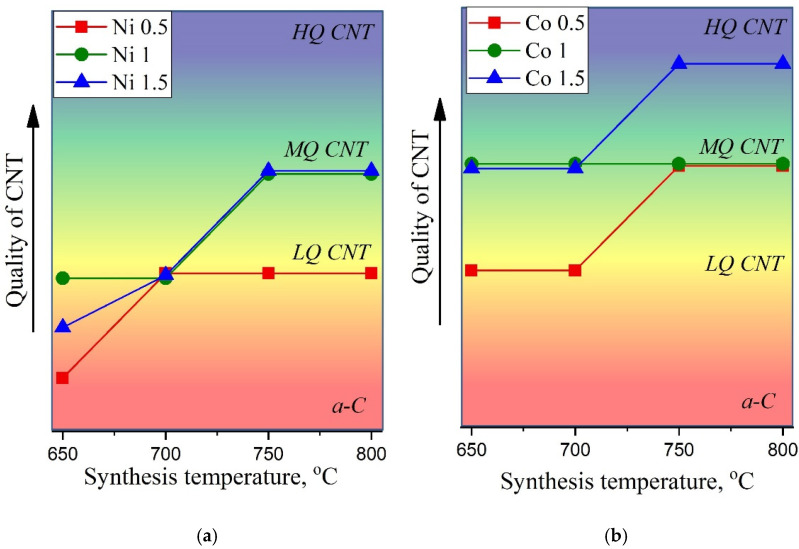
Dependence of the quality of CNTs on the synthesis temperature (**a**) and catalyst concentration (**b**). Where a-C—amorphous carbon, LQ CNT, CNT and HQ CNT—low, medium and high quality of CNT, respectively.

**Figure 10 nanomaterials-12-01817-f010:**
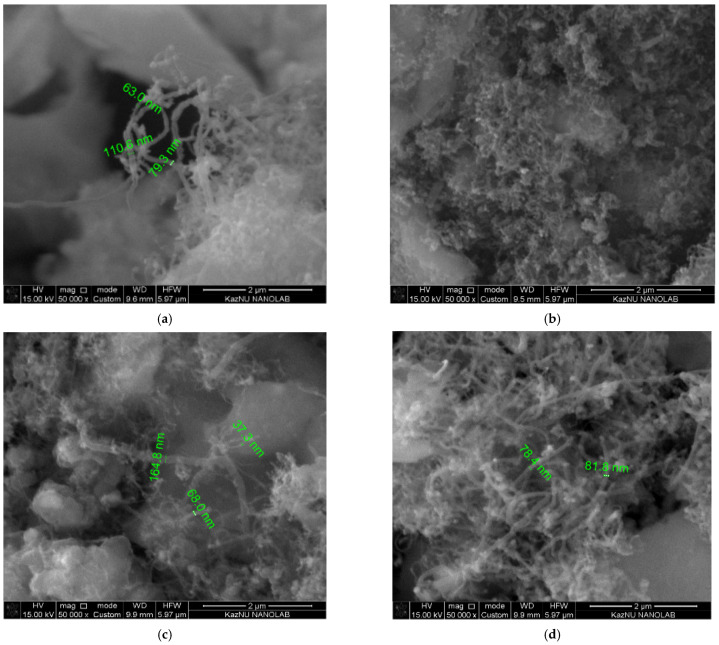
SEM images of MWCNTs. (**a**) T = 750 °C: at (1.5 M; Ni(NO_3_)_2_·6H_2_O); (**b**) T = 800 °C: at (1.5 M; Ni(NO_3_)_2_·6H_2_O); (**c**) T = 750 °C: at (1.5 M; CoCl_2_·6H_2_O); (**d**) T = 800 °C: at (1.5 M; CoCl_2_·6H_2_O).

## Data Availability

Not applicable.

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
