# Peer review of "The Use of Diatomite as a Catalyst Carrier for the Synthesis of Carbon Nanotubes"

_nanomaterials, 2022, doi:10.3390/nano12111817_

Round 1

Reviewer 1 Report

The manuscript describes the synthesis of CNT using the CVD method employing nickel- and cobalt-based catalysers deposited on the surface of the diatomite. The quality of the obtained nanotubes was assessed based on Raman spectrometry.

The issues to consider before publication are listed underneath.

  • ‘For the production of carbon nanotubes using the C-CVD method, the diatomite mineral was used from the Mugaldzhar deposit...’ - Could the authors provide more information about this mineral (for example, XRD spectra, results of chemical analysis, SEM images, etc.) to better inform the reader?
  • ‘We prepared two solutions saturated with Co(NO3)2‧6H2O; Ni(NO3)2‧6H2O, respectively in a 1:1 ratio and dried’ and ‘synthesis temperature for 0.5 M and 1 M concentrations’ – the information is inconsistent. I recommend rearranging the description of the catalyst preparation. In what proportions were the diatomite and catalyst solution mixed (m / m ratio)? What was the concentration of the catalyst solution? Was the concentration of the catalyst solution precisely defined (0,5 or 1 M) or the solution in use was saturated at room temperature?
  • ‘As a result, a metal-containing catalyst (diatomite) was obtained in the form of a powder. Then 0.5 g of the metal-containing catalyst was placed in a reactor and heated in an argon flow to 400-500 °C, the concentration of metal salts in solution with alcohol was equal to 0.5, 1.0 and 1.5 Ðœ of metal salt in solution, the propane-butane mixture served as a carbon carrier’ – why is the solution mentioned here? I assume that the dry, metal-containing catalyst was placed in the oven and then exposed to gaseous reagents.
  • The stoichiometry of R1 needs correction.
  • ‘Dependences of CNT quality on synthesis temperature and catalyst concentration are shown in Figure A 7’ - How is the quality parameter of nanotubes defined? In what units is it expressed? It would be advisable to also add to Figure 8 images of nanotubes obtained under the most favourable conditions when a nickel catalyst.
  • Could the authors rewrite the first paragraph of the conclusions starting with the second sentence? This passage seems a bit confusing in the current version.

Author Response

  • ‘For the production of carbon nanotubes using the C-CVD method, the diatomite mineral was used from the Mugaldzhar deposit...’ - Could the authors provide more information about this mineral (for example, XRD spectra, results of chemical analysis, SEM images, etc.) to better inform the reader?  Answers: We agree with comments and we already add Figure A2. Natural diatomite and SEM analysis of diatomite.
  • ‘We prepared two solutions saturated with Co(NO3)2‧6H2O; Ni(NO3)2‧6H2O, respectively in a 1:1 ratio and dried’ and ‘synthesis temperature for 0.5 M and 1 M concentrations’ – the information is inconsistent. I recommend rearranging the description of the catalyst preparation. In what proportions were the diatomite and catalyst solution mixed (m / m ratio)? What was the concentration of the catalyst solution? Was the concentration of the catalyst solution precisely defined (0,5 or 1 M) or the solution in use was saturated at room temperature? Answer: We agree with comments. So, we reorganized these explanation.
  • ‘As a result, a metal-containing catalyst (diatomite) was obtained in the form of a powder. Then 0.5 g of the metal-containing catalyst was placed in a reactor and heated in an argon flow to 400-500 °C, the concentration of metal salts in solution with alcohol was equal to 0.5, 1.0 and 1.5 Ðœ of metal salt in solution, the propane-butane mixture served as a carbon carrier’ – why is the solution mentioned here? I assume that the dry, metal-containing catalyst was placed in the oven and then exposed to gaseous reagents.  Answer: We have changed this sentences as this:  We prepared two solutions with three different concentration (0.5, 1.0 and 1.5 Ðœ) of Co(NO3)2‧6H2O; Ni(NO3)2‧6H2O, respectively. Then 0.5 gram of diatomaceous earth was impregnated with an alcohol solution of Co(NO3)2‧6H2O; Ni(NO3)2‧6H2O, respectively and dried for 13-15 minutes at a temperature of 80-100 °C, necessary and sufficient for evaporation of water and alcohol. As a result, a metal-containing catalyst (diatomite) was obtained in the form of a powder. Then it was placed in a reactor and heated in an argon flow to 400-500 °C.  In order to synthesis of carbon nanotubes the propane-butane mixture served as a carbon carrier due to its availability. 
  • ‘Dependences of CNT quality on synthesis temperature and catalyst concentration are shown in Figure A 7’ - How is the quality parameter of nanotubes defined? In what units is it expressed? It would be advisable to also add to Figure 8 images of nanotubes obtained under the most favourable conditions when a nickel catalyst.                                        Answer: 

    On the image of CNTs’ quality, we tried to provide a generalizing visual representation of the results of Raman spectra analysis. The quality of nanotubes was assessed by summing up such parameters as FWHM(G), I(D)/I(G), and I(G)/I(2D). The increasing degree of quality is shown by the arrow on the corresponding axis so that it varies from a-C (amorphous carbon) to HQ CNT (high-quality carbon nanotubes). 

  • Could the authors rewrite the first paragraph of the conclusions starting with the second sentence? This passage seems a bit confusing in the current version.  Answer: Yes. We have changed conclusion.

Reviewer 2 Report

Dear Authors,

your article is well written and clear in each parts.

Conclusions are supported by your data. The Conclusions are known. High temperature gives better quality of Carbon Nanotubes and less presence of diffects.

I suggest to the authors to underline the novelty in your work.

I has been difficult to find something new. I found other articles which did Raman analysis to show the quality of the synthesis of Carbon Nanotubes. 

Can authors show the different of their article with previous ones?

Which role does diatomite play on growing of Carbon Nanotubes?

its role is not clear. It is clear the different behaviors of two catalysts Ni and Co.

Do you have take in account the dimension of the catalyst particle on diatomite substrate during the CVD synthesis? This one is an important parameter for the synthesis.

I suggest a major revision of the paper.

Author Response

Diatomaceous earth is often used as a catalyst carrier and as a waste water treatment agent due to its high permeability, high porosity and chemical inertness. 

Their characteristic features are the presence of amorphous active silicon dioxide on the one hand and a fine-porous structure, lightness, and low thermal conductivity on the other. These properties make these materials chemically highly active and make it possible to use them as sorbents, catalysts, filtering and heat-insulating materials and filler carriers. According to the results of Raman spectroscopy, the growth of multi-walled carbon nanotubes was observed on selected catalysts, as Co and Ni at temperatures 750 and 800 ℃.

In addition, heterogeneous systems have lower cost and higher efficiency compared to expensive pure CNTs with better adsorption performance.

It was stated that the optimal conditions of C-CVD process for the growth of high-quality carbon nanotubes were: 1.5 M solution of CoCl2‧6H2O at temperatures of 750 and 800 ℃.

Round 2

Reviewer 1 Report

The authors addressed my comments and added missing information. However, Equation R1 needs correction and its stoichiometry is not correct (2 NO2 molecules are released, not one molecule).

The labels in Figure A3 are indecipherable due to overlap or cutting. 

Author Response

1) We have corrected Equation R1.

2) The labels in Figure A3 also was corrected. 

Reviewer 2 Report

Many Thanks to the authors for they efforts to modify the paper following the reviewer suggestions.

It sounds better and more precise, in this version. I spur the authors to continue to work on this topics considering other important aspects/ parameters which play an important role in the  high quality carbon nanotubes synthesis using CVD technique.

Author Response

Many thanks. Yes, we will continue our work as reviewer suggested in future.